# An Exploratory Analysis of Housing and the Distribution of COVID-19 in Sweden

Mohammad Ismail , Abukar Warsame and Mats Wilhelmsson *

Department of Real Estate and Construction Management, Division of Real Estate Economics and Finance, Royal Institute of Technology (KTH), SE-100 44 Stockholm, Sweden; ismail2@kth.se (M.I.); abukar.warsame@abe.kth.se (A.W.)

* Correspondence: mats.wilhelmsson@abe.kth.se or matswil@kth.se

**Abstract:** The impact of COVID-19 on various aspects of our life is evident. Proximity and close contact with individuals infected with the virus, and the extent of such contact, contribute to the intensity of the spread of the virus. Healthy and infected household members who both require sanctuary and quarantine space come into close and extended contact in housing. In other words, housing and living conditions can impact the health of occupants and the spread of COVID-19. This study investigates the relationship between housing characteristics and variations in the spread of COVID-19 per capita across Sweden's 290 municipalities. For this purpose, we have used the number of infected COVID-19 cases per capita during the pandemic period—February 2020 through April 2021—per municipality. The focus is on variables that measure housing and housing conditions in the municipalities. We use exploratory analysis and Principal Components Analysis to reduce highly correlated variables into a set of linearly uncorrelated variables. We then use the generated variables to estimate direct and indirect effects in a spatial regression analysis. The results indicate that housing and housing availability are important explanatory factors for the geographical spread of COVID-19. Overcrowding, availability, and quality are all critical explanatory factors.

**Keywords:** COVID-19; housing; exploratory analysis

## 1. Introduction

Housing is essential for society and the economy, as well as for individuals and households. The right to adequate housing is vital and represents a base for all economic, social, and cultural rights [1]. According to the UN, adequate housing was recognised as part of the right to an adequate standard of living in the 1948 Universal Declaration of Human Rights and the 1966 International Covenant on Economic, Social, and Cultural Rights [2]. The right to housing enshrined by the UN is not just a right to a basic shelter, but to adequate housing, which includes a set of interlinked factors in terms of the physical structure of the housing. These include affordability, legal security of tenure, the immediate housing environment; the availability of services, facilities, and infrastructure; and the community, i.e., neighbourhood and social relationships, cultural adequacy, and collective efficacy [3,4].

The housing situation is also a critical social determinant of health [5,6]. A shortage of suitable housing and poor housing situations affect health. Nearly 200 years ago, the New York Board of Health wrote in Reports of Hospital Physicians (1832): "The real suffering of the poor is easily explained. They lived in the worst houses in the most crowded portions of the city and could not afford to flee when threatened by the epidemic".

COVID-19 has significantly affected the world's economy, demographics, and human behaviour [7]. The housing sector was important during the pandemic, which brought extended periods of lockdowns [8–10]. During lockdowns, people were forced to do all activities at home. The lengthy periods indoors created various health, physical, psychological, and social challenges, as well as challenges associated with dwelling characteristics,

such as maintaining physical distance in apartment buildings where lobbies, elevators, or laundry facilities are shared. Housing thus became a focus during the pandemic [11].

The first case of COVID-19 in the Nordic region was in northern Finland on 29 January 2020, and the second two days later in Sweden. A month later, the virus began spreading in the Nordic countries, mainly from individuals returning from winter holidays in Northern Italy. In Sweden, the second case was confirmed on 26 February. Thereafter, the virus began to spread mainly in Stockholm and at the largest Swedish skiing resort in the Jämtland Härjedalen region, with the Stockholm region accounting for around 40% of all national cases by the end of April [12].

Sweden suffered during the early phase of the epidemic compared to the rest of Scandinavia, with very many people infected and a high number of deaths, especially among those over 70 living in elderly care facilities. The measures introduced by Sweden were less strict than other Nordic countries and focused on guidelines and recommendations instead of imposing restrictions and lock-downs, with the measures being voluntary rather than compulsory to a high degree [12–14].

The Public Health Agency of Sweden introduced a set of recommendations such as encouraging individuals to avoid public transport, crowding, social gatherings, and all non-essential travels, including that between regions; to work from home when possible, to make arrangements to secure social distancing, and to move to distance teaching in high schools and universities [12].

With the start of the epidemic in 2020, data showed a rise in mortality and morbidity from COVID-19 among immigrants in Scandinavian countries. Until the mid-20th century, Denmark, Norway, and Sweden had a relatively similar immigration history. Since the 1970s, Sweden has received the largest share of refugees per capita in Europe, reflected in the proportion of immigrants in Scandinavian countries: 19.6% in Sweden, 14.7% in Norway, and 10% in Denmark. Given the large number of infections and deaths, Sweden represents the most substantial evidence of disparities in the effect of COVID-19 by country of birth in Scandinavian countries. Immigrants suffer from poorer socioeconomic conditions in terms of lower incomes and lower employment rates, is the gap in employment rates between immigrants and the rest of the population reaches 25% in Sweden. In addition, residential segregation occurs, where immigrants live in immigrant-dense neighbourhoods with a high overall population density [14].

Socioeconomic status (education, income, and employment status), number of working-age household members, and neighbourhood population density explain up to half of the increased COVID-19 mortality risks among migrants in Stockholm [15]. Sigurjónsdóttir et al. (2021) examined the factors relating to increased risk of COVID-19 infection in Scandinavian countries at the city-district level in socio-economically vulnerable, low-income districts in Oslo, Helsinki, Copenhagen, Stockholm, and Malmö [12]. The studied factors were annual income, the share of residents with foreign backgrounds, the share of inhabitants working in exposed occupations, and overcrowded housing conditions. In addition, in in-depth studies at the sub-district level in Stockholm and Malmö, the study used more variables, such as household sizes, educational level, car ownership, and spatial density. The results indicate a higher spread of COVID-19 among the population with foreign backgrounds, with low-income earners and vulnerable immigrants suffering most. Place of birth, spatial segregation, socioeconomic inequalities between neighbourhoods, and housing conditions impact the daily lives of this population and, depending on where they live, can increase the risk of being infected with COVID-19 [12].

Similarly, Florida and Mellander (2020) studied the geographic factors associated with the spread of COVID-19 in Sweden. They found an association between the geographic variation of COVID19 cases and factors such as density, population size, socioeconomic characteristics, household size, and predominance of elderly homes. However, they concluded that all these variables provide little explanation for the variation of COVID-19 across Sweden [13].

Our objective is to investigate the connection of the spread of COVID-19 per capita in Sweden's municipalities with housing and the housing situation in these municipalities. Hence, the main research question we seek to answer is: does a relationship exist between housing and the spread of COVID-19 in Sweden's municipalities? That is, our purpose is to analyse the impact of housing conditions on the spread of COVID-19 and not the impact COVID-19 has had on the housing market.

Our main contribution to the research is empirical. Unlike previous studies, we focus on the importance of the housing situation to general health and the spread of COVID-19 in particular. Admittedly, many of the relationships that appear are not causal, but it is nevertheless interesting to make them visible. The focus has been on the importance of housing, but a picture emerges of the importance of socioeconomic factors, showing vulnerability and clarifying how society should prepare to manage future crises.

There are several reasons that vaccination rates could differ between two groups of people: vaccine accessibility, vaccination priority based on risk level, vaccine acceptance, and hesitancy due to historical or cultural reasons [16]. A recent study in Sweden shows that a large portion of the population (around 20%) firmly say no to getting vaccinated [17]. A noticeable limitation in this paper that would have improved our analysis is the availability of vaccinated and unvaccinated persons in different municipalities. We also did not discuss various communication channels that health agencies and municipalities utilised in sharing information related to COVID-19.

The rest of the paper is divided into five main sections. Section 2 provides a literature review for the importance of housing and its relationship to COVID-19 and health. Section 3 describes the different data sets used and explains our research methodology. Section 4 presents the description of the empirical results. Section 5 discuss the results, and the last section includes conclusions drawn from the study.

## 2. Literature Review

The World Health Organisation [18] emphasises that housing conditions and quality have significant implications for peoples' health and can reduce disease, save lives, and increase quality of life. In addition to its role in achieving several Sustainable Development Goals, including good health and wellbeing (SDG 3) and sustainable cities and communities (SDG 11), housing is important because the world's urban population is predicted to double by 2050, with commensurate increases in demand for housing in developing and developed countries.

Figure 1 illustrates the importance of housing for individuals and households. The home protects the individual in the form of shelter. Housing that meets the most basic requirements also enables physical, mental, and social wellbeing. The individual feels that they are part of a community, which is vital for feeling well. The essential factors that create belonging are associated with home, memory, familiarity, and local social relations. In addition to the continuity of residence as a determining factor for belonging to a neighbourhood, housing also provides security, privacy, and a perceived sense of safety. People's overall happiness or quality of life correlates with housing satisfaction and conditions. Housing characteristics and types significantly impact residential satisfaction [19,20]. Housing is more than just shelter; it also fulfils several human social needs. Housing satisfaction can also be viewed as an indicator of individual happiness and wellbeing [21,22].

The community-health aspects of housing go back to the social reformers of the 19th century [23]. The link between poor housing and poor health was established as early as 1842, in Edwin Chadwick's 'Report into the Sanitary Conditions of the Labouring Population of Great Britain' [24]. John Snow (1855) indicated the impact of housing on people's health and wellbeing by providing statistical evidence of the relationship between residence and cholera infection in London [25].

This still holds today. During the COVID-19 pandemic, housing played an essential role in the progression of the pandemic and responses to it [26]. Housing is a significant so-

cial determinant of health [27]. One of the most important inequality dimensions impacting the pandemic is housing [28].

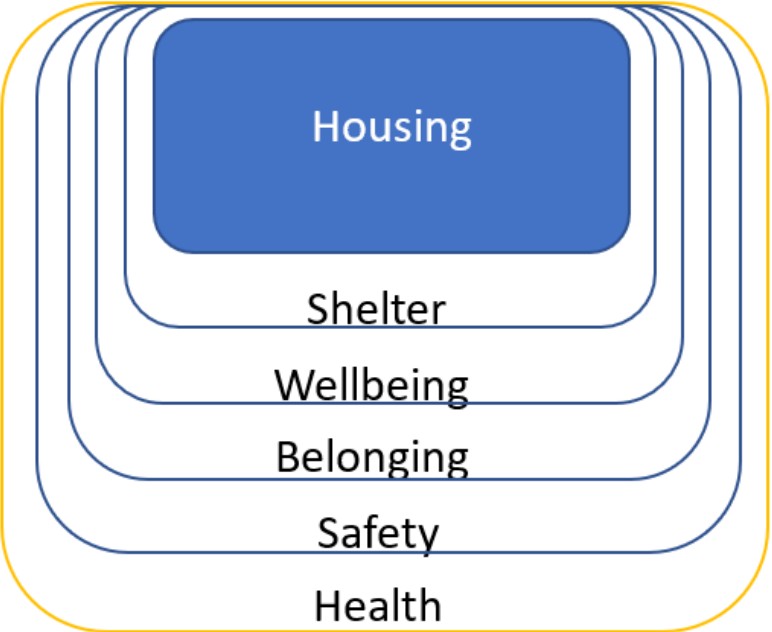

**Figure 1.** The importance of housing.

There is a wide-ranging and growing body of knowledge about health and housing [29,30]. Taylor (2018) classifies research on housing's impact on health into four main pathways [31].

First, the 'stability' pathway concentrates on housing insecurity's physical and mental health effects, including significantly higher morbidity [32]. Pollack et al. (2004) examined the association between housing conditions in terms of tenure and self-rated health for a sample of the population in Germany. The study provides evidence of a significant relationship between house ownership and self-rated health in Germany. People living in rented homes have poor self-rated health [33].

Second, the 'safety and quality' pathway focus on the health effects of conditions inside the house, where environmental factors such as inadequate ventilation, high humidity, and residential crowding, which represent the most frequently used indicators of housing conditions, are all correlated with poor health [24]. Many studies indicate the possibility of improving health through enhancing housing quality. Jacobs et al. (2009) analysed the relationship between health status and housing quality over time by studying housing and health trends from about 1970 to 2000 for the US population and its housing. The results indicated consistent relationships between housing conditions and certain health trends over time [29]. Bonnefoy (2007) refers to the housing environment and conditions as significant factors affecting human health [34]. Moreover, housing conditions can be related to poorer health outcomes and the spread of infectious diseases [35].

Third, the 'affordability' pathway describes the health effects of unaffordable housing. Bailey (2020) emphasises the association between housing and health and the role of affordable housing in supporting people's health. High housing costs worsen low-income families' health and forces them to make difficult choices between spending on essential needs—such as medicines, food, and heating—and paying rent. The absence of affordable housing obliges families to live in housing or neighbourhoods that lack proper health and safety conditions, increases the number of family members living in one house together, and worsens adults and children's physical and mental health conditions [36].

Fourth, the 'neighbourhood' pathway focuses on the health effects of the environmental and social characteristics of the neighbourhoods in which the people live. Neighbourhood segregation, crime rates, and social capital affect health [31].

Our study can be characterised as combining some of the housing characteristics described in the pathways above. Low-income households living in rental apartments with high rents might be subject to overcrowding, contributing to poor health and the spread of diseases such as COVID-19.

## 3. Data and Methods: Explanatory Analysis

The exploratory analysis will consist of several steps. The theoretical starting point is that the geographical spread of COVID-19 is affected by many factors. How people live and how their homes are shaped is an essential piece of the puzzle in understanding the spatial spread of COVID-19. First, we present the dependent variable in the analysis. This, of course, refers to the number of COVID-19 cases per capita in 290 municipalities in Sweden in 2020/2021. Then, several underlying independent control variables are included in the analysis, such as population characteristics and socioeconomic factors. The main focus of this study is the variables that measure housing and housing conditions in the municipality. We use a large number of variables that are quite highly correlated. Therefore, we begin the exploratory analysis by analysing the correlations between included variables. Then, we perform a principal component analysis, where a number of new variables are created that consist of all components included in the analysis, but in which the new variables are not correlated. In the last step of the exploratory analysis, these new variables are used to estimate direct and indirect effects in a spatial regression analysis.

### 3.1. Variables

The variables consist of aggregated data at the municipal level. The sources for the data are Statistics Sweden and the Swedish Public Health Agency. The number of COVID-19 cases per municipality refers to the pandemic period, February 2020 through April 2021. Data regarding population and socioeconomic factors refer to the most recent available year, usually 2019. This also applies to the factors that measure the housing situation in the municipalities. Table 1 below illustrates descriptive statistics regarding the number of COVID-19 cases per 100,000 inhabitants and the variables that describe the housing situation in the municipalities.

**Table 1.** Descriptive Statistics.

| Variable | Obs. | Mean | Std. Dev. | Min. | Max. |
|---|---|---|---|---|---|
| COVID-19 | 290 | 653.617 | 888.72 | 167 | 9711 |
| Housing Stock | 290 | 17,166.341 | 37,266.457 | 1140 | 497,690 |
| Elderly Housing | 290 | 476.817 | 863.339 | 24 | 11,587 |
| Condominium | 290 | 10 | 9.105 | 0 | 63.2 |
| Single-Family | 290 | 65.46 | 13.565 | 3.3 | 87.9 |
| Rental | 290 | 23.388 | 7.651 | 4.4 | 47.7 |
| Adults at Home | 290 | 40.623 | 10.269 | 16 | 77.5 |
| Housing Prices | 290 | 2264.369 | 1663.097 | 330 | 11,209 |
| Segregation | 290 | 22.956 | 8.42 | 1.6 | 46.8 |

Note. Variable definitions: COVID-19: cases per capita (infected cases/10,000 inhabitants) during the pandemic period—February 2020 through April 2021—per municipality. Housing Stock: number of houses per municipality. Elderly Housing: quantity of elderly housing, per municipality. Condominium: percentage housing in condominium, per municipality. Single-Family: percentage housing in single-family houses, per municipality. Rental: percentage housing in tenancy, per municipality. Adults at Home: percentage of adults living with parents, per municipality. Housing Prices: house price in thousand Swedish kronor (SEK), per municipality. Segregation: the segregation index measures the difference in settlement patterns between different population categories. The calculation is made here for those born abroad, where 0 means even distribution over the area and 100 means total segregation.

### 3.1.1. COVID-19

The variation in COVID-19 cases between Swedish municipalities is considerable, from only 167 up to 9700 cases per 100,000 inhabitants. The mean number of COVID-19 cases was over 653, with a standard deviation of 889. Of course, the large spread is explained because the municipalities have different total and elderly populations sizes.

Figure 2 illustrates the spread of COVID-19 cases per municipality. The redder the municipality, the more COVID-19 cases reported. Some geographical clusters have more cases of COVID-19. The three metropolitan regions, for example, have more cases per 100,000 inhabitants compared with many other cities. The purpose of the exploratory analysis is to investigate whether the housing situation in the municipality can explain this geographical variation.

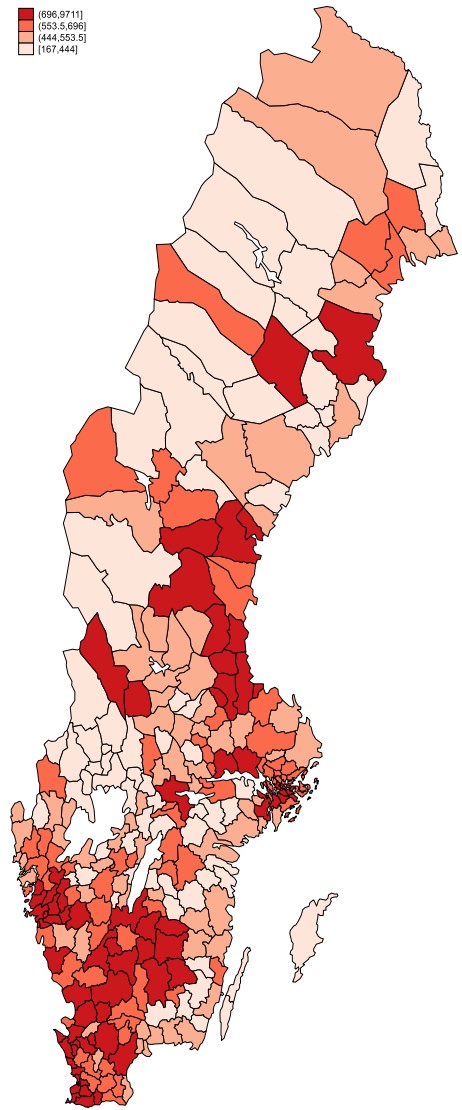

**Figure 2.** Distribution of COVID-19 cases across municipalities.

The distribution of COVID-19 cases between the municipalities in Sweden is random, as partially stated earlier in the literature, but we hypothesise that variables that measure the housing situation in the municipality play an essential role. The housing variables included in the analysis are described below.

### 3.1.2. Housing Characteristics

Our theoretical starting point is that housing and living conditions impact health and the spread of COVID-19. We analyse several variables, all of which are intended to quantify housing in the municipality in different dimensions, namely, (1) housing stock, (2) special housing for the elderly, (3) type of tenure, (4) housing situation, and (5) segregation.

First, we relate the number of dwellings to the number of residents in the municipality. The fewer homes, the higher the housing density on average. The number of dwellings is also related to the municipality's area. Higher housing density can have a negative impact on the spread of COVID-19. In other words, there is a positive correlation between housing density and COVID-19 cases. The variation in size between the 290 municipalities is considerable. On average, the municipalities have just over 17,000 homes, but the standard deviation is as much as 37,000. The smallest municipality has only 1140 homes, while the largest (Stockholm) has 498,000 homes.

Second, the amount of elderly housing per capita varies between the municipalities. An increased proportion of special housing for the elderly probably negatively impacts the spread—the average number of elderly housing units per municipality is 478. However, the variation is considerable between the municipalities, with only 24 units in the municipality with the lowest number of units and 11,587 in the municipality with the most. This variation is also a result of population variation across municipalities in general, and specifically the variation of the population over 85.

Third, another group of variables intends to measure the tenure form of housing in the municipality. Here, we use the ratio of the number of dwellings in multi-family houses to single-family houses. The hypothesis is that a higher proportion of dwellings in multi-family houses increases the spread. We have also included the number of owner-occupied housings related to the total number of dwellings. Again, there is no clear hypothesis, although it can be expected that homeowners have a higher average income and a lower housing density, which have influenced the spread of the virus. All the housing tenure variables interact with demographics, education, and place of birth variables in our extended models. On average, the housing of Sweden's population is divided among 10% condominiums, 65% single-family homes, and about 23% rental stock. However, the variation between Sweden's municipalities is significant. There is a complete lack of condominiums in some municipalities, while in others, almost two-thirds are condominiums. We can observe the most significant variation within the group of single-family houses, between 3% and 88%. The proportion of residents in rental housing also varies considerably.

Fourth, we have also included variables intended to measure how "difficult" the housing situation is in the municipality. We measure this, among other things, by comparing housing prices to the average income of municipality inhabitants. We have also included the proportion of households that cohabitate with adult children. We hypothesise that the housing situation in the municipality has meant that the spread of COVID-19 has been greater than would otherwise have been expected. Two measures of housing market strain are used. Both measure housing affordability in some form. House prices indicate how limited the housing supply is in relation to demand in the local housing market. Of course, demand is affected by many factors, such as income and other socioeconomic factors, but housing prices are undoubtedly a function of both demand and supply. Low supply in relation to demand means that prices are higher. On average, the price of a single-family house in Sweden is SEK 2.2 million, with a standard deviation of as much as SEK 1.7 million. The cheapest single-family house costs just SEK 300,000 and the most expensive SEK 11 million. The second variable used to measure affordability is the proportion of households with adult children (over 18 years of age) cohabitating. On average, that proportion amounts to 40%, with a variation of 16 to 78%.

Finally, we measure how housing-segregated the municipality is in terms of ethnic background. This variable's expected effect on the spread of COVID-19 is not apparent, but a cautious hypothesis could be that increased segregation of low-income households' harms health and thus has a positive impact on the spread of COVID-19. The segregation

index measures the proportion of the population that must move to achieve a completely even distribution of all residents in the municipality. The variation is noticeable from only 1.6% to 46.8%, with an average of 23%, i.e., a quarter of the population must move in order for us to achieve an even ethnic distribution in the municipality. The measure is most suitable for analysing segregation within a municipality over time and not comparing municipalities. Figure 3 below illustrates the correlation between the housing variables used in the analysis.

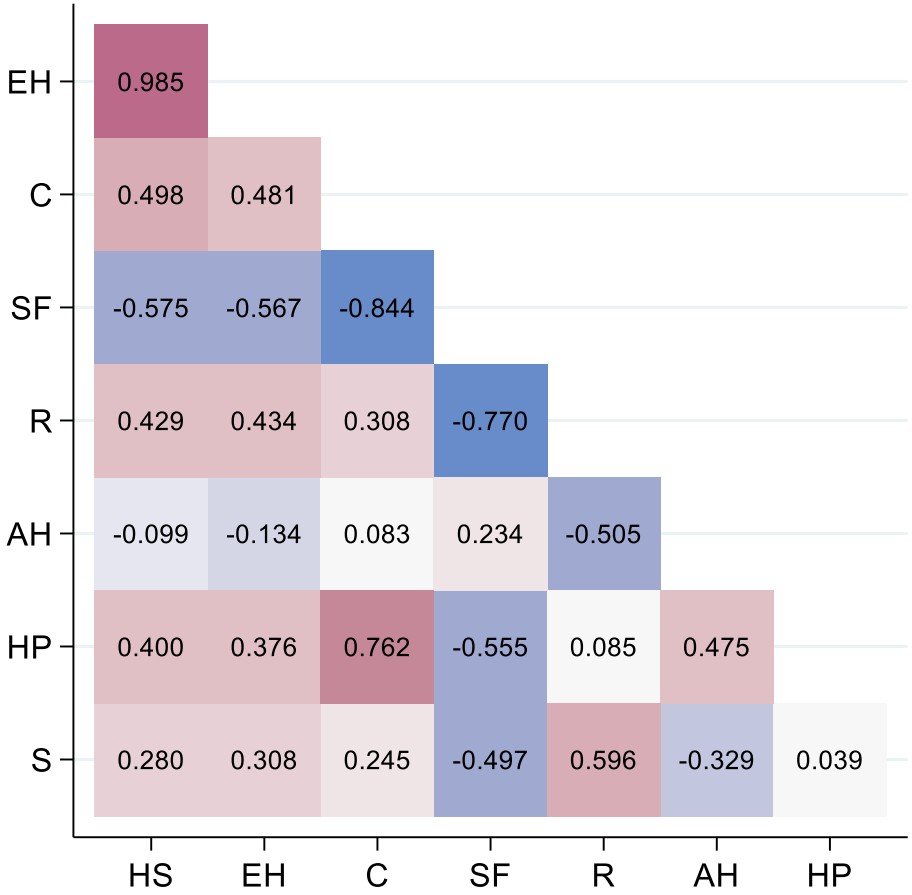

**Figure 3.** Correlation coefficients (housing variables). Note: Elderly Housing (EH), Condominium (C), Single-Family (SF), R(Rental), AH (Adults at Home), HP (House prices), and S (Segregation).

The correlation between the number of dwellings in the municipality and the number of elderly dwellings is significant, with a positive correlation coefficient of 0.98. We can also note that the correlation between housing prices and the proportion of condominiums is positive and relatively high (0.76). The proportion of single-family houses is unsurprisingly negatively correlated (−0.84) with the proportion of condominium housing, and the shares of rental apartments and single-family houses are also highly negatively correlated (−0.77). We can also note a high negative correlation between housing prices and single-family houses.

The correlation indicates that it will be difficult to draw any conclusions if all variables explain the spread of COVID-19 between Sweden's municipalities. Therefore, we have chosen to use a method called Principal Component Analysis. What we do, in principle, is group the variables that have a high mutual correlation and create so-called components or factors. These constructed components will be mutually uncorrelated by definition and can be used as independent variables in the regression models.

### 3.1.3. Control Variables

Housing variables impact the spread of COVID-19 between the municipalities, but several underlying factors are also important, as indicated by, for example, Florida and Mellander (2021) [13]. These include the age distribution in the municipality, the proportion of migrants, income level, and level of education. We include these as control variables in the exploratory analysis and these variables interact with the housing variables. The higher the average age in the municipality, the more people are hypothesised to be infected with COVID-19. The proportion of migrants and the education level can affect the spread, as migrants and the low-skilled may be less informed about protecting themselves. These groups may also be more likely to drive buses or taxis, clean, work in elderly care, or do other jobs significantly affected by the pandemic. Increased income is thus expected to reduce the spread of COVID-19 cases.

## 4. Results

As mentioned earlier, we are using Principal Component Analysis (PCA) to analyse the housing and control variables. PCA is a statistical transformation to convert a set of correlated variables into linearly uncorrelated variables, enabling us to reduce the dimensions while retaining most of the information and identifying the most influential variables [37]. PCA was performed with one as the Eigenvalue threshold value. Table A1 in Appendix A illustrates the PCA results for the housing variables, and Table A2 in Appendix A illustrates the PCA results for all housing and the interaction variables between the housing and control variables.

As Table A1 illustrates, two factors were extracted, each with Eigenvalues above one, explaining 74% of the total variance. The analysis shows that about 50% of the total variation is explained by the first principal component and 24% by the second component. Table A2 also shows the factor loadings for the variables, where the variables with factor loadings lower than 0.60 indicate that those variables do not fit the factor solution well and should possibly be dropped from the analysis [38].

The components *hh1* and *hh2* can be described based on the factor loadings. The component hh1 measures the total housing stocks, housing price, and non-owner-occupied houses, which can be hypothesised to be positively related to COVID-19. This component can be categorised as a combination of the stability and affordability pathways. The component *hh2* includes mostly variables indicating the degree of the populations living in rental apartments in segregated areas (the neighbourhood pathway), which can also be hypothesised as positively related to COVID-19. However, unexpectedly, the proportion of households with adult children at home has a negative loading.

Two tests were used to test the data's suitability for Principal Component Analysis. The Kaiser–Meyer–Olkin (KMO) measures the sampling adequacy for each variable and the complete model. The test value is between 0 and 1, and a KMO test value of more than 0.5 is considered adequate [39]. As Table 2 shows, the KMO test value is 0.591. The second test is Bartlett's Test of Sphericity ($X2 = 3707.978$, $p = 0.00$), where a significant *p*-value of less than 0.05 indicates that Principal Component Analysis is adequate and valid for our data.

**Table 2.** Range of component.

| Variable | Obs. | Mean | Median | Std. Dev. | Min. | Max. | Range |
|---|---|---|---|---|---|---|---|
| *hh1* | 290 | 0 | −0.293 | 1 | −1.129 | 7.835 | 8.963 |
| *hh2* | 290 | 0 | 0.0666 | 1 | −3.647 | 2.415 | 6.062 |
| *h1* | 290 | 0 | −0.147 | 1 | −0.996 | 13.236 | 14.233 |
| *h2* | 290 | 0 | −0.171 | 1 | −2.141 | 4.775 | 6.916 |
| *h3* | 290 | 0 | −0.1374 | 1 | −2.053 | 4.214 | 6.268 |
| *h4* | 290 | 0 | 0.0673 | 1 | −3.384 | 2.286 | 5.6697 |
| *h5* | 290 | 0 | −0.0412 | 1 | −3.457 | 4.131 | 7.588 |

Table A2 illustrates five extracted factors, each with Eigenvalues above one, explaining 81.31% of the total variance. The analysis shows that about 41% of the total variation is explained by the first principal component, 18% by the second component, and around 12% by the third component—together, 71% of the cumulative explained variance for the first three components. The last two components explain about 10% of data variance. Regarding the measurement of sampling adequacy, the Kaiser–Meyer–Olkin (KMO) value is $0.855 > 0.5$, and Bartlett's Test of Sphericity ($X2 = 14626.423$, $p = 0.00 < 0.05$). This indicates that Principal Component Analysis is adequate and useful for our data.

The components *h1–h5* can be described based on the factor loadings. The component *h1* comprises older people and is expected to positively relate to COVID-19 infection per capita (socioeconomic factors). The component *h2* can be characterised as younger individuals with higher educations and living in owner-occupied dwellings with children (stability and affordability pathways). The expected relationship to COVID-19 cases is positive. In Sweden, kindergartens and schools for children aged 6–16 remained open during the pandemic, with few exceptions. According to the assessment of Public Health Sweden, closing all schools in Sweden was not an effective measure. This decision was based on analysing the situation and the potential consequences of school closures on Swedish society. Thus, the expected relationship to COVID-19 cases is positive, with schools remaining open during the pandemic. Component *h3* measures the degree of unemployed persons born abroad and primarily living in rental apartments (stability pathway), and is positively related to COVID-19. Component *h4* measures the housing quality and percentage of adult children living with parents (affordability and quality pathway); the higher the housing quality and the fewer adult children living with parents, the fewer COVID-19 cases expected. Component *h5* measures the geographical size of the municipality (neighbourhood pathway); larger size and less density should result in lower COVID-19 cases.

Table 2 presents the descriptive statistics regarding the components focusing on their rank. The first two components used in Model 1 (baseline model) are *hh1* and *hh2*. By definition, the mean value will be equal to 0 and the standard deviation will be equal to 1. However, we can note that component *hh1* has a skewed distribution towards positive values, while *hh2* is negative. The component *hh1* shows a higher variation across municipalities than component *hh2*. The range for *hh1* is equal to 9, while for *hh2*, it is around 6. In Model 2 (extended model), we will use the components *h1–h5*. The first three components have a positive skew, while the last two have a negative skew. It is primarily component *h1* that shows considerable variation across municipalities. The range for *h1* is equal to 14.23, indicating that variation is high. The other components have much less variation, around half for *h2–h5* compared to *h1*. Figures 4 and 5 below illustrate the spatial distribution of components *hh1* and *hh2*, and components *h1–h5*, respectively.

By definition, the components are not correlated. Nevertheless, one can see relatively clear patterns in geography. Housing component *hh1* (blue map) has a concentration in the metropolitan regions and municipalities in central Sweden also have positive values for the component *hh1*. This does not apply as clearly to component *hh2* (red), which is concentrated in Sweden's central and southern parts.

Component *h1* (red) is mainly concentrated along the coasts of Sweden, including the metropolitan regions, while component *h2* (green) is more spatially concentrated in municipalities in and around the metropolitan regions. Component *h3* (blue), on the other hand, is primarily concentrated in the middle and southern parts of Sweden, excluding the metropolitan regions. Component *h4* (purple) is predominantly concentrated in central Sweden, excluding the metropolitan regions, and component *h5* (grey) is primarily concentrated in northern Sweden's interior.

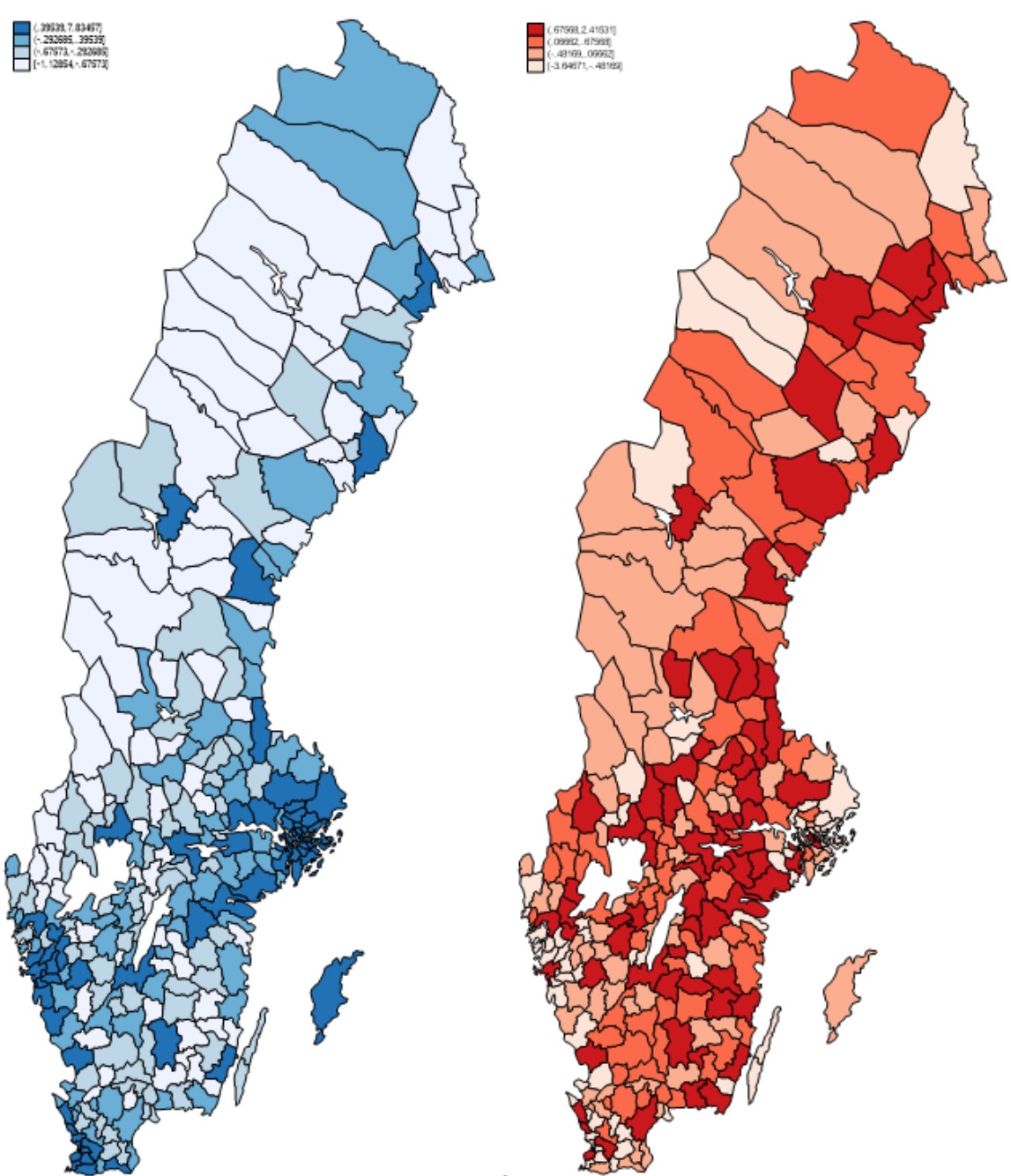

**Figure 4.** Housing components (*hh1–hh2*) distribution in space.

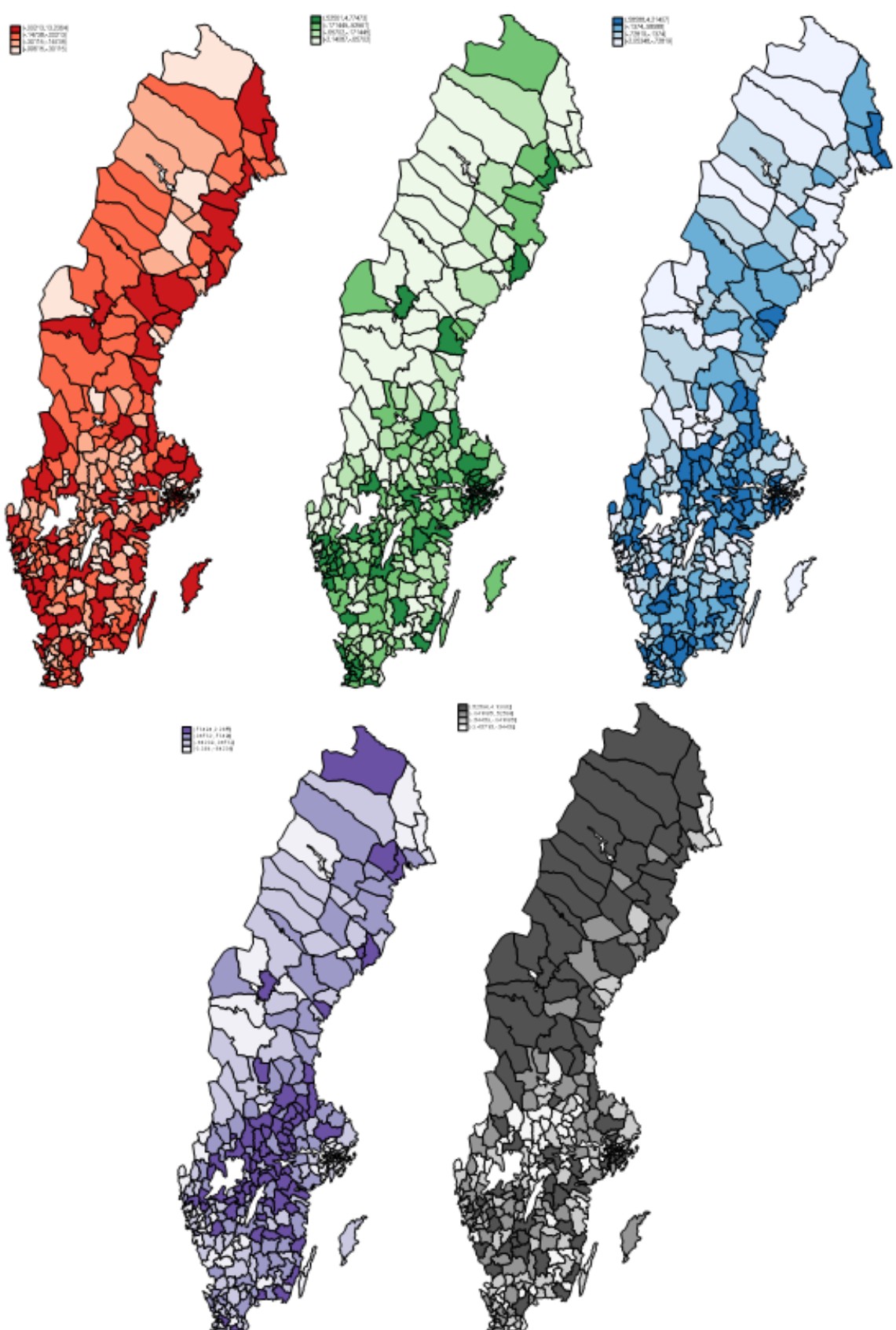

**Figure 5.** Housing components (*h1–h5*) distribution in space.

## 5. Discussion

In order to examine the relationship between COVID-19 cases per capita and the variables created through Principal Component Analysis (PCA) for the housing and control variables, we have estimated a linear regression model; the results are presented in Table 3. In the regression model, housing and control variables were considered explanatory variables, and COVID-19 cases per capita were the dependent variable.

**Table 3.** Regression Results.

| Variable | (1) Baseline | (2) Extended |
|---|---|---|
| *hh1* | 527.9 *** | |
| | (12.86) | |
| *hh2* | 165.6 *** | |
| | (4.03) | |
| *h1* | | 737.1 *** |
| | | (26.55) |
| *h2* | | 67.38 * |
| | | (2.43) |
| *h3* | | 117.2 *** |
| | | (4.22) |
| *h4* | | −62.36 * |
| | | (−2.25) |
| *h5* | | −74.39 ** |
| | | (−2.68) |
| Constant | 653.6 *** | 653.6 *** |
| | (15.95) | (23.59) |
| *N* | 290 | 290 |
| $R^2$ | 0.388 | 0.723 |
| *AIC* | 4623.9 | 4399.9 |

Note: *t* statistics in parentheses; * $p < 0.05$, ** $p < 0.01$, *** $p < 0.001$.

The explanatory variables explain approximately 39% of the variation in the dependent variable in Model 1 (baseline model). All the explanatory variables are statistically significant with expected signs. Component *hh1* is significantly stronger than component *hh2*, in that the estimated parameter has a significantly higher magnitude. The standard deviation of parameter estimation is also more minor than the coefficient compared to *hh2*. That is, housing stock and housing prices (affordability pathway) seem more important than rental housing market size and degree of segregation (neighbourhood pathway).

In Model 2 (extended model), the degree of explanation is much stronger. Almost 72% of the variation between Sweden's municipalities in the spread of COVID-19 can be explained. Moreover, the variables that describe the housing situation are significantly more convincing when interacting with the background variables. Component *h1* has the greatest impact and is statistically significant at a 1% statistical level (socioeconomic and demographic factors). Hence, the size of the housing market and special housing for the elderly can explain a large portion of COVID-19 cases per capita and municipality. This also applies to component *h3*, which measures the number of foreign-born persons, unemployment, and rental apartments (socioeconomic factors and the stability pathway). However, all components are significant at a 5% level. As expected, *h1–h3* positively affect the spread of COVID-19, while *h4* and *h5* have a negative effect (affordability and quality pathway). Thus, all these effects are in line with expectations.

We have also analysed whether the parameter estimates are constant between the municipalities and the municipality types. We have partly analysed large and small markets by analysing the municipalities with fewer homes than average, compared with those with more than average. We have also analysed the municipalities with low and high housing prices. The results of these analyses can be found in Table 4.

**Table 4.** Parameter heterogeneity.

| Variable | (1) Smaller | (2) Larger | (3) Less Expensive | (4) More Expensive |
|---|---|---|---|---|
| h1 | 101.1 | 778.0 *** | 81.64 | 762.1 *** |
|  | (1.24) | (12.53) | (1.37) | (16.25) |
| h2 | 118.3 *** | 70.00 | 132.8 *** | 85.22 |
|  | (6.49) | (0.62) | (5.47) | (1.07) |
| h3 | 64.82 *** | 307.6 ** | 78.54 *** | 188.2 ** |
|  | (4.79) | (3.31) | (5.17) | (3.03) |
| h4 | −26.18 * | 59.93 | −18.36 | −90.59 |
|  | (−2.25) | (0.56) | (−1.08) | (−1.31) |
| h5 | 15.46 | −381.2 * | 23.35 | −212.2 * |
|  | (1.44) | (−2.46) | (1.84) | (−2.36) |
| Constant | 599.3 *** | 487.2 * | 605.0 *** | 567.4 *** |
|  | (24.42) | (2.52) | (28.03) | (4.97) |
| N | 222 | 68 | 186 | 104 |
| $R^2$ | 0.264 | 0.790 | 0.234 | 0.762 |
| AIC | 2870.9 | 1114.4 | 2412.3 | 1669.1 |

Note: $t$ statistics in parentheses; * $p < 0.05$, ** $p < 0.01$, *** $p < 0.001$.

The analysis of parameter heterogeneity is interesting and very clear. In smaller and more affordable municipalities, *h2* and *h3* (the affordability, neighbourhood, and stability pathways) play a role in spreading COVID-19. At the same time, in larger regions, components *h1* and *h3* (socioeconomic and demographic factors and the stability pathway) are the prominent explanatory factors. It is also clear that the degree of explanation differs markedly between the smaller and larger municipalities. In smaller localities with lower housing prices, only about one-quarter of the spread of COVID-19 can be explained by the housing variables, but in larger municipalities where housing prices are higher, almost 80% of the variation can be explained. The spread of COVID-19 in smaller municipalities with lower housing prices can be explained by other things not included in our models, and the housing and the other independent variables included can explain a large part of the spread in larger municipalities. Hence, in the smaller municipalities, there is a risk that we have excluded variables that should be included and that this may have created an omitted variable bias.

Spatial data tends to be spatially dependent, caused by either spatial heterogeneity or spatial dependence. Therefore, we have analysed the results by estimating spatial autoregressive (SAR) models and spatial error models (SEM). We use the inverse distance between the municipalities as a spatial weight matrix (row standardised). The results from these models are presented in Table 5.

The spread of COVID-19 between the municipalities does have spatial dependence. The results from the SAR and SEM models are partly different in that all parameter estimates are not statistically significant. Similar to the OLS estimates, the models in which the interaction variables are included better explain the variation in the spread; Models 3 and 4 have greater explanatory power than Models 1 and 2, according to AIC. Component *h1* clearly still best explains the variation in the dependent variable. The other components have a lower significance level, with components *h4* and *h5* differing from zero at a 5% level of statistical significance in Model 4.

**Table 5.** Spatial autoregressive and error model.

| Variable | (1) SAR | (2) SEM | (3) SAR | (4) SEM |
|---|---|---|---|---|
| *hh1* | 640.8 *** | 605.0 * | | |
| | (3.32) | (2.46) | | |
| *hh2* | 70.50 | 151.4 *** | | |
| | (1.33) | (3.41) | | |
| *h1* | | | 739.4 *** | 736.1 *** |
| | | | (7.48) | (7.46) |
| *h2* | | | 38.92 | 67.45 |
| | | | (1.00) | (1.93) |
| *h3* | | | 112.6 | 121.2 |
| | | | (1.80) | (1.92) |
| *h4* | | | −44.70 | −50.53 * |
| | | | (−1.84) | (−2.28) |
| *h5* | | | −66.27 * | −72.89 * |
| | | | (−2.35) | (−2.57) |
| Constant | 1806.7 ** | 693.7 ** | 405.4 | 649.1 *** |
| | (3.16) | (2.96) | (1.95) | (9.29) |
| Rho | −1.692 * | | 0.364 | |
| | (−2.08) | | (1.29) | |
| Sigma | 669.9 *** | 686.9 *** | 465.5 *** | 464.0 *** |
| | (5.11) | (5.29) | (3.96) | (4.01) |
| Lambda | | 0.688 | | 0.561 * |
| | | (1.57) | | (2.11) |
| *AIC* | 4611.6 | 4623.4 | 4402.4 | 4401.1 |

Note: $t$ statistics in parentheses; * $p < 0.05$, ** $p < 0.01$, *** $p < 0.001$.

## 6. Conclusions

Housing characteristics and health factors associated with human wellbeing may seem obvious in our daily life. Nonetheless, after the COVID-19 pandemic, a better understanding of the potential outcomes of their interactions is necessary. Proximity and prolonged contact between household members could contribute to the spread of the viruses in households and subsequently in the broader community, including workplaces, schools, and houses of worship where individuals interact outside the home. Numerous studies have sought to establish the extent and the direction of any perceived relationship between housing and health factors. The contributions of numerous essential factors to household wellbeing are already well established. However, the simultaneous role of housing as a sanctuary, providing comfort, privacy, and security as well as minimising the spread of the COVID-19, has not been previously explored.

The main focus of this study is to ascertain the existence and the extent of any perceived relationship between housing and the spread of COVID-19 in Sweden's municipalities. First, several variables concerning housing and housing conditions that could capture availability, crowding, and quality effects in each municipality were collected and analysed. Not surprisingly, many of the variables, especially those associated with housing stocks and tenures, were highly correlated with each other. Including all the variables in the analysis of possible relation to the spread of COVID-19 would make drawing any reliable conclusions difficult. Thus, exploratory and principal component analysis (PCA) allowed us to generate a set of uncollated relevant variables suitable to estimate the direct and indirect effects of the spread of COVID-19 in different municipalities in Sweden.

To a certain degree, our results provide a plausible explanation of how the combination of various housing characteristics and socioeconomic factors would have helped or hindered the spread of COVID-19 in Sweden. The PCA's two main components, measuring housing stocks and housing tenures and the degree of the population living in rental apartments in segregated areas, indicate spatial dependence and a positive relationship to COVID-19. Similarly, the other five components obtained from the PCA (based on

control variables) indicate geographical patterns related to the spread of the COVID-19. For instance, a high number of COVID-19 cases were reported on the coast of Sweden, where major cities are located, and a high concentration in older people is evident.

Furthermore, based on parameter heterogeneity analysis, our results show that housing stocks and house prices in different regions play a role in the spread of COVID-19. In other words, housing availability and affordability contribute to how different municipalities experienced the pandemic during the study period. Concerning the spatial distribution aspect, the component representing the number of elderly houses and older persons best explains the variation of COVID-19 cases observed in different municipalities. To a lesser extent, two other components, representing housing quality and geographical size in different municipalities, are statistically significant and thus seem to contribute to the spread of COVID-19.

A couple of important suggestions could be derived from our exploratory analysis of housing and the distribution of COVID-19. Although this study was carried out at the municipal level rather than specific areas of each municipality, the availability of more affordable and better housing qualities in most vulnerable areas would create an environment where housing stocks can fulfil both the sanctuary and quarantine needs during crises such as the COVID-19 pandemic. These improvements in the quality and affordability of housing might also contribute to the wellbeing of households living in the so-called miljonprogram, who experience overcrowding and other socioeconomic challenges. Elderly housing is another sector that requires comparative assessment since the older population was the segment of society that experienced the most significant share of COVID-19 cases.

Preparedness for health crises such as COVID-19 and the implementation of policies and strategies to minimise socioeconomic consequences necessitate the allocation of necessary resources and prior knowledge of vulnerable groups in the society that might not be easily observed before a pandemic occurs. The policy implication of this study is that municipalities should strive to provide proper housing to curtail the spread of diseases in vulnerable groups such as low-income households and the elderly population.

One of housing's main characteristics is an intrinsic connection with the surrounding neighbourhood. Thus, further investigation and more focused analysis of specific areas in different municipalities could yield meaningful relationships between housing conditions and the overall wellbeing of households.

**Author Contributions:** Conceptualisation, M.W.; methodology, analysis, initial draft preparation, M.I., A.W. and M.W.; writing—review and editing, A.W. and M.W. All authors have read and agreed to the published version of the manuscript.

**Funding:** This research was funded by Housing 2.0 (Bostad 2.0).

**Institutional Review Board Statement:** Not applicable.

**Informed Consent Statement:** Not applicable.

**Data Availability Statement:** The source for the data used in research is the Swedish Public Health Agency for the number of COVID-19 cases per municipality https://www.folkhalsomyndigheten.se/the-public-health-agency-of-sweden/ accessed on 23 April 2021. Data regarding population and socioeconomic factors are available on Statistics Sweden https://www.scb.se/ accessed on 23 April 2021.

**Acknowledgments:** We thank the research project Housing 2.0 (Bostad 2.0) for financial support.

**Conflicts of Interest:** The authors declare no conflict of interest.

## Appendix A

**Table A1.** Principal Component Analysis (PCA) for the housing variables.

| Variables | hh1 | hh2 |
|---|---|---|
| Condominium | **0.886** | 0.019 |
| Housing Prices | **0.856** | −0.363 |
| Single-Family | **−0.803** | −0.469 |
| Housing Stock | **0.753** | 0.319 |
| Elderly Housing | **0.734** | 0.350 |
| Adults at Home | 0.228 | **−0.845** |
| Rental | 0.375 | **0.807** |
| Segregation | 0.255 | **0.682** |
| **Eigenvalue** | 3.995 | 1.936 |
| **Explained Variance (%)** | 49.943 | 24.206 |
| **Cumulative Explained Variance (%)** | 49.943 | 74.149 |
| **Kaiser–Meyer–Olkin Measure of Sampling Adequacy** | 0.591 | |
| **Bartlett's Test of Sphericity** | | |
| Approx. Chi-Square | 3707.978 | |
| Sig | 0.000 | |

**Note:** bold numbers are factor loadings higher than 0.60.

**Table A2.** Principal Component Analysis (PCA) for housing and interaction variables.

| Variables | h1 | h2 | h3 | h4 | h5 |
|---|---|---|---|---|---|
| Number of people 95+ years | **0.971** | 0.185 | 0.081 | 0.043 | −0.012 |
| House stock total | **0.969** | 0.208 | 0.095 | 0.028 | 0.028 |
| Number of people 65–74 | **0.964** | 0.230 | 0.099 | 0.038 | 0.043 |
| Number of elderly houses | **0.963** | 0.199 | 0.079 | 0.066 | 0.054 |
| Number of people 85–94 | **0.959** | 0.235 | 0.106 | 0.068 | 0.051 |
| Low income <160,000 SEK | **0.957** | 0.210 | 0.140 | 0.035 | 0.041 |
| Number of people 75–84 | **0.949** | 0.267 | 0.113 | 0.045 | 0.056 |
| House price _single family house | 0.257 | **0.837** | −0.038 | −0.283 | −0.151 |
| % Share with post-secondary education, born in Sweden | 0.350 | **0.828** | −0.145 | −0.109 | −0.013 |
| % Housing in condominium | 0.313 | **0.807** | 0.233 | 0.040 | −0.052 |
| % Share with post-secondary education, foreign born | 0.204 | **0.789** | −0.295 | −0.169 | −0.132 |
| % People living in urbanised areas | 0.169 | **0.763** | 0.297 | 0.055 | −0.098 |
| % Share 0–19 year, born in Sweden | 0.031 | **0.660** | 0.314 | −0.495 | −0.027 |
| % Housing in own house | −0.391 | **−0.649** | −0.515 | −0.261 | −0.151 |
| % Share 0–19 year, foreign born | −0.220 | **−0.627** | −0.095 | 0.282 | −0.070 |
| % Share 20–64 year, foreign born | 0.269 | **0.615** | 0.105 | −0.197 | 0.517 |
| % Share 20–64 year, born Sweden | 0.343 | 0.573 | −0.198 | 0.289 | 0.287 |
| % Foreign born | 0.152 | 0.257 | **0.829** | −0.155 | −0.107 |
| % long-term unemployed, age 20–64 | 0.126 | 0.176 | **0.818** | 0.027 | −0.080 |
| % Registered unemployed, age 20–64 | −0.014 | −0.415 | **0.744** | 0.338 | −0.014 |
| % Age 20–64 neither gainfully employed nor studying | 0.024 | −0.380 | **0.655** | 0.106 | −0.331 |
| % Housing in tenancy | 0.320 | 0.203 | **0.635** | 0.407 | 0.332 |
| Segregation index | 0.199 | 0.135 | 0.489 | 0.302 | 0.389 |
| % Adult children living with parents | −0.101 | 0.223 | −0.164 | **−0.800** | −0.269 |
| Older housing stock | 0.109 | −0.184 | 0.193 | **0.786** | −0.231 |
| Municipality area | 0.001 | −0.279 | −0.200 | 0.034 | **0.647** |
| **Eigenvalue** | 10.676 | 4.711 | 3.088 | 1.626 | 1.039 |
| **Explained variance (%)** | 41.061 | 18.119 | 11.876 | 6.255 | 3.995 |
| **Cumulative explained variance (%)** | 41.061 | 59.179 | 71.055 | 77.310 | 81.305 |
| **Kaiser–Meyer–Olkin Measure of Sampling Adequacy** | | | 0.855 | | |
| **Bartlett's Test of Sphericity** | | | 14,626.423 | | |
| Approx. Chi-Square | | | 0.000 | | |
| Sig | | | | | |

Note: bold numbers are factor loadings higher than 0.60.

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
