# Peer review of "An Exploratory Analysis of Housing and the Distribution of COVID-19 in Sweden"

_buildings, doi:10.3390/buildings12010071_

Round 1
Reviewer 1 Report
Given the significant impact of COVID-19 on the global society, economy, and individual well-being, exploring the correlation between housing characteristics and the spread of COVID-19 is an important and timely study. Based on 290 municipalities in Sweden from Feb. 2020 to April. 2021, this paper empirically analyzes the impact of housing conditions on the spread of COVID-19, and finds that housing quality and availability are critical to explaining the geographical spread of COVID- 19.
But there are several concerns regarding the method and findings of the paper. First of all, the panel data adopted in this paper can test the dynamic correlation between variables. This data characteristic is essential for the current research because it is well known that the spread of COVID-19 is a contagious process. The correlation between this change and dynamic process and housing characteristics should be more meaningful and effective in revealing the importance of housing corresponding to health. However, the current paper is not based on panel data methods.
Second, the spread of COVID-19 depends mainly on the degree of spatial separation of people, partly due to the living density mentioned by the authors but partly is due to people’s behavior. A simple example might be that people in the community realize the importance of keeping distance, thereby strengthening self-protection, leading to a low transmission of COVID-19. I understand this might be difficult to measure, but more discussion on this issue is very important. It needs to confirm why the paper findings are not biased if this critical factor is ignored.
Table 6 finds that for “smaller” and “less expensive” cities, the R-square is very low, indicating that the variables included in the regression cannot explain the spread of COVID-19 in these areas. More variables that are important for understanding the effect of COVID – 19 of these municipalities are missing. It cannot be simply due to the degree of randomness pointed out by the authors. More discussions are necessary.
Some min comments are introduction is rather long in the current version, 1.2 should be a separated section on the literature review; several figures are not necessary or could be moved to the appendix, such as Figure 1, Figure 5. Table 3 is also suggested to move to the appendix.

Author Response
Many thanks for the comments.
(1) do not know if we have been unclear but we do not have panel data which makes it impossible to use panel data models.
(2) We have added a paragraph about vaccination:
"There are several reasons that vaccination rates could differ between two groups of people: vaccine accessibility, vaccination priority based on risk level, vaccine acceptance, and hesitancy due to historical or cultural reasons [16]. A recent study in Sweden shows that a large portion of the population (around 20%) firmly say no to getting vaccinated [17]. A noticeable limitation in this paper that would have improved our analysis is the availability of vaccinated and unvaccinated persons in different municipalities. We did not also discuss various communication channels that health agencies and municipalities utilized in sharing information related to COVID-19. "
(3) Absolutely correct. Here we were very vague. We have added a paragraph about
"The spread of COVID-19 in smaller municipalities with lower housing prices can be explained by other things not included in our models, and the housing and the other independent variables included can explain a large part of the spread in larger municipalities. Hence, in the smaller municipalities, there is a risk that we have excluded variables that should be included and that this can create an omitted variable bias. "
(4) We have separated the first section into two sections.
(5) We have moved Tables 2 and 3 to the appendix.
Reviewer 2 Report
Review report for the paper “An exploratory analysis of housing and the distribution of COVID-19 in Sweden”. The topic is interesting and the paper is well structured.
In suggest the following improvements to be done:
Insufficient expression on innovative explanations. Does the practical significance of this paper exist? There is a lack of comparison with previous studies of the same kind. For this point, the innovativeness of the author's statement needs further explanation.
Indicator issues. How you have defined the indicators?
Literature review. Add more recent papers published in last three years. Remove papers published before 2017. Based on the LR you should define the scientific gap.
Add limitations and advantages of the proposed study and methodology.
The author has poorly discussed the results of the paper. One would expect to find the previous empirical work enriching the discussions of the results, but unfortunately, that has not been done.
The part of the recommendations is rather short, maybe you can strengthen that part in a way which really show the implication of the findings more clearly.
Author Response
Many thanks for the comments.
(1) We have added a paragraph at the end
"Preparedness of health crises like COVID-19 and implementation of policies and strategies to minimize socioeconomic consequences necessitates the allocation of necessary resources and prior knowledge of vulnerable groups in society that might not be easily observed before a pandemic occurs. The policy implication of this study is that municipalities should strive in providing proper housing to curtail the spread of diseases to vulnerable groups such as low-income households and the elderly population. "
We have also added a longer section to the introduction
"The early spreading of COVID-19 in the Nordic Region was in Northern Finland on 29 January 2020, and the second two days later in Sweden. A month later, the virus began spreading in the Nordic countries, mainly by individuals returning from winter holidays in Northern Italy. In Sweden, the second case was confirmed on 26 February. Thereafter, the virus began to spread mainly in Stockholm, and the largest Swedish skiing resort in the Jämtland Härjedalen region, where the Stockholm region accounted for around 40% of all national cases the end of April [12].
Sweden suffered during the early phase of the epidemic compared to the rest of Scandinavia, with very many infected and a high number of deaths, especially among those over 70 who live in elderly care facilities [14].
The measures introduced by Sweden were less strict than other Nordic countries and adopted guidelines and recommendations instead of imposing restrictions and lockdowns, which were to a higher degree voluntary rather than compulsory [12,13].
The Public Health Agency of Sweden introduced a set of recommendations, such as encouraging individuals to avoid public transport, crowding, social gatherings, and all non-essential travels, also between regions. Work from home when possible. Make arrangements to secure social distancing, and move to distance teaching in high schools and universities [12]. "
(2) We have inserted a note under Table 1 that defines the variables we have used.
(3) We have not found any relevant articles that should be included in the literature review. We have also included relevant articles even if they were published before 2017.
(4) all results are discussed based on the pathways we presented in our theoretical starting point. Then there are no studies, as far as we know, that have empirically shown the relationship between the housing situation and the spread of COVID-19, which makes it difficult to relate our results to these.
Round 2
Reviewer 1 Report
The authors have carefully addressed all my comments. no more comments.
Reviewer 2 Report
The authors have addressed the point of my concern. I am happy with their corrections. Hence, I would like to recommend this manuscript to be published.